# Impact of Virtual Reality Simulation in Endodontics on the Learning Experiences of Undergraduate Dental Students

Raidan Ba-Hattab [1,*], Dilek Helvacioglu-Yigit [1], Lamyia Anweigi [1], Tayeb Alhadeethi [1], Mahwish Raja [1], Sundus Atique [1], Hanin Daas [1], Rebecca Glanville [2], Berkan Celikten [3], Kaan Orhan [3,4] and Kamran Ali [1]

1   College of Dental Medicine, QU Health, Qatar University, Doha P.O. Box 2713, Qatar
2   Faculty of Health, Plymouth University, Plymouth PL4 8AA, UK
3   Faculty of Dentistry, Ankara University, Ankara 0600, Turkey
4   Medical Design Application and Research Center (MEDITAM), Ankara University, Ankara 0600, Turkey
*   Correspondence: rbahattab@qu.edu.qa

**Abstract:** We aimed to evaluate the impact of Virtual Reality Dental Simulators (VRDS) on preclinical training in endodontics for undergraduate dental students. Purposive sampling technique was used to target undergraduate dental students at two dental schools: in Qatar and Turkey. After training on endodontic access cavity preparation on upper anterior teeth using acrylic teeth on dental mannequins and virtual reality haptic dental simulator, a questionnaire based on a combination of open- and closed-ended items was distributed to the participants. The sample included 60 dental undergraduate students. The participants reported positive experiences with VRDS and 76% considered it to be helpful in improving their fine motor skills. Endodontic access cavity preparation on VRDS was perceived to be similar to natural and acrylic teeth by 73.34% and 53% of participants, respectively. Overall, 85% of participants supported the use of VRDS training to supplement conventional training on dental mannequins but also recommended the need for improvements in VRDS training in endodontics. The findings of this study underscore the benefits of VRDS in endodontics. Future research involving larger samples from multiple institutions may help to optimize VRDS in undergraduate dental education.

**Keywords:** dental; endodontics; simulation; students; virtual reality

## 1. Introduction

Undergraduate dental students are expected to develop competence in a range of operative procedures to enable them to deliver safe and effective clinical dental care [1]. Dentistry involves performing invasive procedures which may cause irreversible harm to the patients. A fundamental strategy to ensure safe clinical care by dental students is to provide preclinical training in simulated dental learning environment. Simulated settings provide a safe learning environment and are particularly important for training and assessments of students irreversible dental procedures before they undertake these procedures on real patients in clinical settings [2]. Development of competence in practical skills in operative dental procedures requires acquisition of fine motor skills involving hand–foot and eye coordination. Historically, preclinical training has been largely based on learning operative dental skills on extracted natural teeth as well as plastic teeth (typodonts), which are mounted as models on mannequins [3]. This approach is well-established in most dental schools globally and has been used for several decades. However, training on mannequins entails a running expenditure on materials with limited opportunities for repeated practice by the students. Moreover, it requires availability of appropriately qualified supervisors and potential issues of inter-rater reliability during assessments of dental students [4,5].

Dental virtual reality simulators with haptic feedback are becoming increasingly popular in dental schools and can potentially address some of the limitations of endodontic

teaching on mannequins. The available systems can evaluate the learning process and provide immediate feedback with minimal supervisor dependence [4]. Apart from haptic feedback during the exercise, students have the opportunity to practice repeatedly without any additional cost implications [6,7]. Moreover, students receive immediate, three-dimensional, audio and written feedback on their work on artificial teeth (such as a cavity, crown, and endodontic access preparations). In addition, they may be able to review their work following completion as a playback video file [8].

Several studies have been conducted to test the validity of Virtual Reality (VR) technology [6,9,10]. Previous work by Buchanan has shown that stakeholders view VR simulators positively and students trained on VRDS learned faster, demonstrated the same level of performance, accomplished more practice procedures per hour, and requested more evaluations than in the traditional preclinical laboratories [9]. Jasinevicius et al., reported that the quality of the student preparations when comparing students using only traditional instruction and those who used the VRDS are comparable [10]. Studies have also reported that VR technology improves student performance, and allows them to obtain higher examination and course scores [8–10]. There is a growing trend of using VR in a preclinical dental curricula as VRDS training has a positive impact on students' ergonomic development and technical performance [8,9,11–13].

Evidence from the literature suggests that teaching endodontics to undergraduate dental students presents multiple challenges for dental educators [14] Most students view this discipline as complex and difficult to master, which in turn reflects on their skills and confidence in performing root canal treatment, especially on multi-rooted teeth [15]. Competency in endodontics is widely recognized as one of the more challenging skills for undergraduate dental students, as reported in multiple studies [16,17].

A variety of strategies have been employed in preclinical endodontic teaching, including a comparison of training on artificial or natural teeth; use of apex locator during preclinical training in endodontics; and evaluation of the technical quality of root canal treatment performed by the undergraduate students [18,19]. Studies have also evaluated the effectiveness of VRDS training with haptic feedback and using microcomputed tomography images from real patient cases to simulate natural tooth anatomy in VRDS. The results showed that the use of VRDS allows a more accurate tooth preparation with a lower risk of removal of healthy tooth structure [20]. Additionally improved speed of access opening, better bimanual dexterity and better force utilization following training on haptic VR has also been reported [21]. Feedback from dental students shows that they value the integration of VRDS into the endodontic curriculum as it allows improved comprehension of root canal anatomy [22]. However, further understanding is required on the interplay between the conventional endodontic teaching on mannequins and VR and how its impacts on the learning experiences of undergraduate dental students.

The aim of this study was to explore the impact of VRDS in preclinical endodontics on the learning experiences of undergraduate dental students.

*Conceptual Framework*

Advances in medical education have influenced educators to view health-care education through the lens of sociology and anthropology in order to capture all the influences and interactions that transpire in the learning environments through active engagement of the learners [23]. The conceptual framework of the study is underpinned by the theory of Situated learning [24]. It asserts that learning is a transformative process and always intricately tied to its context and to the social relations and practices therein. It emphasizes the social nature of cognition, and the importance of authentic situations and activities to facilitate learning [25].

The dental profession represents a fraternity who come together in pursuit of a shared enterprise. The role of a student may be viewed as one of legitimate peripheral participation [24]. The newcomers begin learning at the periphery of the dental community, initially by observing and later by performing basic tasks. As they become more knowl-

edgeable and skilled, they move centrally. Through participation, active engagement and assuming increasing responsibility, the newcomers acquire the roles, skills, and values of the culture and community. As learners are transformed through participation and social interaction with the community, they in turn exert an influence to transform the community. In the context of the present study, VRDS offers a safe learning environment to introduce dental students to complex operative procedures, i.e., endodontics through peripheral participation and facilitates development of competence for a smooth transition into clinical practice.

## 2. Material and Methods

### 2.1. Research Ethics

Ethical approval was obtained from the Institutional Review Board Qatar University (Reference number: QU-IRB 1652-EA/22).

### 2.2. Study Setting

This was a multi-institution study at Qatar University (Qatar) and Ankara University (Turkey).

### 2.3. Study Design

Interventional study to explore the impact of virtual reality dental haptics in preparation of endodontic access cavities.

### 2.4. Sampling Technique and Participants

Purposive sampling was used to recruit undergraduate students in third year of the undergraduate dental programs in Qatar and Turkey. Invites to participate in the research were sent to potential participants by email. The invites were accompanied by a participant information sheet explaining the purpose and scope of the study.

### 2.5. Research Instrument

A questionnaire consisting of ten closed- and four open-ended items was drafted by the research team consisting of five experienced clinical dental academics (Appendix A). The questionnaire was aimed at investigating the learning experiences of undergraduate students enrolled on a course in endodontics. The questionnaire explored the perceptions of the participants in endodontic access cavity preparation using both conventional methods based on endodontic teeth mounted on models as well as VRDS. Face and content validity of the items was determined through mapping of the questionnaire items to the learning outcomes of endodontic curriculum related to access cavity preparation.

The development of the questionnaire was aimed at eliminating biases related to person factors adopting the following steps:

- Recall of relevant skills and behaviors were easy for the respondents,
- The items were worded in a clear and unambiguous manner to facilitate easy and consistent interpretation of items by the participants,
- The questionnaire was administered online to allow participants to provide their responses anonymously. It also meant that the researcher could not influence their responses which avoided a halo effect.

Having generated the potential items for inclusion in the questionnaire, pretesting of items was conducted in line with established practices. Undergraduate dental students (N = 5), and dental academics (N = 5) participated in pretesting of the questionnaire electronically. The purposes of pretesting of the scale items were as follows:

1. Determine the content and face validity of scale items.
2. Determine the clarity and consistent interpretation of the questionnaire by the participants.
3. Determine clarity of the scoring categories.

### 2.6. Data Collection

The participants at both institutions were provided traditional training on endodontic access cavity preparation on natural and acrylic teeth mounted on physical models in mannequins. Additionally, the students received training on endodontic access cavity preparation on VRDS. Student at Qatar University used SIMtoCare (Vreeland, The Netherlands) VRDS while in Turkey the students use Simodont VR (Nissin Dental Products, Kyoto, Japan). Both VRDSs provided haptic feedback. Maxillary central incisors were used for all endodontic exercises to facilitate consistency in comparisons between training using traditional methods and VRDS at both institutions.

Following completion of the training, the participants were invited to complete an online questionnaire using google forms. Prior to accessing the questionnaire, each participant was required to sign an electronic consent form to confirm they understood the purpose of the study; their participation was voluntary; and that the data would be processed anonymously.

### 2.7. Data Analyses

All data were analyzed using RStudio (version 2022.02.3+492) incorporating R version 4.0. Descriptive statistics including confidence intervals were calculated for each school and for the combined dataset. A one-way Analysis of Variance was used to determine any significant variation between the institutions' results.

### 3. Results

A total of 60 dental students responded to the questionnaire, with 14 from School A (Qatar) and 46 from School B (Turkey). The quantitative data related to ten closed-ended items with a five-point Likert scale were analyzed first followed by responses to open-ended items.

The responses to closed ended items on the questionnaire were assigned a numerical code as follows:

- Strongly agree: 2
- Agree: 1
- Unsure: 0
- Disagree: −1
- Strongly Disagree: −2

A percentage count of average scores to each item are summarized in Table 1 including responses by school as well as combined responses.

**Table 1.** Participants' responses on pre-clinical training in endodontics.

| Item | | Institution | Strongly Agree | Agree | Unsure | Disagree | Strongly Disagree |
|---|---|---|---|---|---|---|---|
| 1. | The case presentation on VR allowed me to clearly comprehend the tasks expected from me | A | 42.86% | 50.00% | 7.14% | 0.00% | 0.00% |
| | | B | 45.65% | 41.30% | 10.87% | 0.00% | 2.17% |
| | | Combined | 45.00% | 43.33% | 10.00% | 0.00% | 1.67% |
| 2. | The hardness and texture of models in the VR felt similar to natural teeth | A | 21.43% | 42.86% | 21.43% | 14.29% | 0.00% |
| | | B | 28.26% | 47.83% | 17.39% | 14.29% | 0.00% |
| | | Combined | 26.67% | 46.67% | 18.33% | 6.67% | 1.67% |
| 3. | The hardness and texture of models in the VR felt similar to artificial teeth used for endodontic training | A | 7.14% | 21.43% | 7.14% | 42.86% | 21.43% |
| | | B | 15.22% | 45.65% | 21.74% | 10.87% | 6.52% |
| | | Combined | 13.33% | 40.00% | 18.33% | 18.33% | 10.00% |

**Table 1.** *Cont.*

| Item | | Institution | Strongly Agree | Agree | Unsure | Disagree | Strongly Disagree |
|---|---|---|---|---|---|---|---|
| 4. | The images of the teeth and instruments on the VR accurately simulated real structures | A | 0.00% | 64.29% | 7.14% | 0.00% | 0.00% |
| | | B | 0.00% | 60.87% | 6.52% | 2.17% | 0.00% |
| | | Combined | 0.00% | 61.67% | 6.67% | 1.67% | 0.00% |
| 5. | Access cavity preparation on VR was easier compared to acrylic teeth | A | 7.14% | 14.29% | 21.43% | 28.57% | 28.57% |
| | | B | 26.09% | 39.13% | 21.74% | 13.04% | 0.00% |
| | | Combined | 21.67% | 33.33% | 21.67% | 16.67% | 6.67 |
| 6. | The training exercise on the VR improved my fine motor skills | A | 28.57% | 42.86% | 14.29% | 7.14% | 7.14% |
| | | B | 32.61% | 50.00% | 15.22% | 2.17% | 0.00% |
| | | Combined | 31.67% | 48.33% | 15.00% | 3.33% | 1.67% |
| 7. | VR training improved my confidence in learning Endodontics | A | 28.57% | 35.71% | 7.14% | 21.43% | 7.14% |
| | | B | 43.48% | 32.61% | 19.57% | 4.35% | 0.00% |
| | | Combined | 40.00% | 33.33% | 16.67% | 8.33% | 1.67% |
| 8. | VR can replace the preclinical endodontics training on models | A | 7.14% | 21.43% | 7.14% | 42.86% | 21.43% |
| | | B | 17.39% | 30.43% | 19.57% | 4.35% | 10.87% |
| | | Combined | 15.00% | 28.33% | 16.67% | 26.67% | 13.33% |
| 9. | I would like to have more VR sessions in Endodontics | A | 21.43% | 35.71% | 21.43% | 7.14% | 14.29% |
| | | B | 52.17% | 41.30% | 6.52% | 0.00% | 0.00% |
| | | Combined | 45.00% | 40.00% | 10.00% | 1.67% | 3.33% |
| 10. | VR Training may be used to supplement pre-clinical training on mannequin models | A | 42.86% | 50.00% | 0.00% | 7.14% | 0.00% |
| | | B | 28.26% | 56.52% | 13.04% | 0.00% | 1 2.17% |
| | | Combined | 31.67% | 55.00% | 10.00% | 1.67% | 1.67% |

The results were positive overall, as depicted in Table 2. The lowest score was observed for Question 8 'VR can replace the preclinical endodontics training on mannequin models'.

**Table 2.** Mean scores (combined) with 95% Confidence Intervals.

| Question | Mean | StDev | LowerCI | UpperCI |
|---|---|---|---|---|
| 1 | 1.3 | 0.79 | 1.1 | 1.5 |
| 2 | 0.9 | 0.93 | 0.66 | 1.14 |
| 3 | 0.28 | 1.21 | −0.03 | 0.6 |
| 4 | 1.2 | 0.63 | 1.04 | 1.36 |
| 5 | 0.47 | 1.2 | 0.16 | 0.78 |
| 6 | 1.05 | 0.87 | 0.82 | 1.28 |
| 7 | 1.02 | 1.03 | 0.75 | 1.28 |
| 8 | 0.05 | 1.31 | −0.29 | 0.39 |
| 9 | 1.22 | 0.94 | 0.97 | 1.46 |
| 10 | 1.13 | 0.79 | 0.93 | 1.34 |
| Overall | 0.86 | 1.07 | 0.59 | 1.14 |

*3.1. Comparison between Participating Schools*

Participants from School B was more positive about endodontic training on VRDS, with statistically significant variation by in question responses by Institution (when controlling for variation between questions) ($p \leq 0.001$). The findings are depicted in Figure 1.

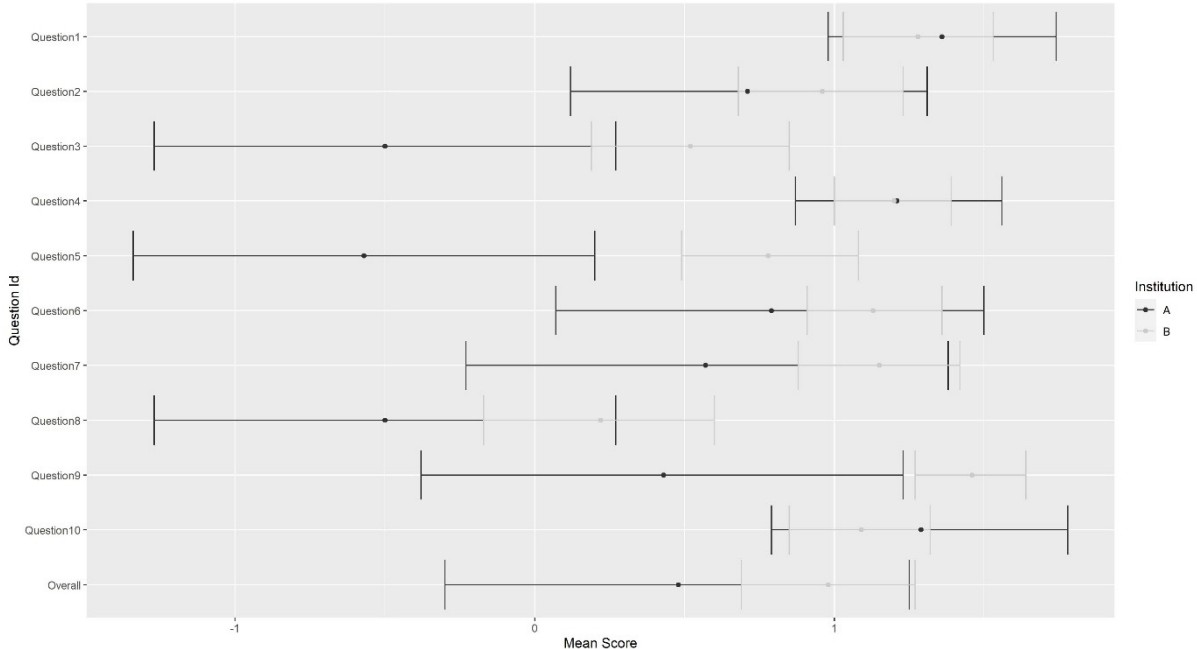

**Figure 1.** Mean score and confidence intervals by question and school.

*3.2. Responses to Open-Ended Items*

A thematic analysis of responses to open-ended items was conducted. The key strengths and limitations of VRDS in Endodontics are summarized in Table 3.

**Table 3.** Benefits and Limitations of Virtual Reality Simulation in Endodontics.

| Benefits | Limitations |
|---|---|
| Opportunities for repeated practice +++ | Lack of proper vision ++ |
| Mimics natural tooth sensation/feeling +++ | Differ from natural teeth (e.g., hardness, drop in feeling, dimension) +++ |
| Improved motor skills ++ | Insufficient VR session time + |
| Improved understanding of dental structures (pulp chamber, root canals) ++ | Difficulty in controlling handpiece ++ |
| Indirect practice + | Adaptation is difficult, needs more practice ++ |
| Improved confidence, decrease stress + | Limited opportunities to apply dental materials + |

+ (1–30%), ++ (31–60%), +++ (More than 60%).

Finally, the participants also suggested improvements in VRDS in Endodontics which are summarized in Table 4.

**Table 4.** Recommendations for Future Development of Virtual Reality Dental Simulation in Endodontics.

| Participants' Recommendations |
|---|
| • It should start before pre-clinical practice on mannequins, preferably in Year 2. |
| • Diverse types of root canal anatomical models might be introduced into the software. |
| • Software should be improved (e.g., the negotiation of root canal orifices, irrigation of root canals, and water running through air motor handpiece are not available) |
| • Automated feedback is insufficient, additional feedback by the facilitator might be helpful |

## 4. Discussion

Learning practical skills in endodontics is considered to be one of the most challenging tasks in undergraduate dental education [26], primarily due to marked anatomical variations in natural teeth. Access cavity preparation is a fundamental skill in endodontics and

requires an accurate balance between conservation of healthy tooth structure and preparing a cavity which is suitable to gain a straight-line access. Preclinical training in endodontics most is frequently delivered using acrylic and/or natural teeth using mannequin models.

Difficulty in negotiating the complex root canal anatomy, particularly in multi-rooted teeth, has been reported as a common challenge by undergraduate students [27]. The introduction of VRDS has offered an innovative option to visualize root canal anatomy and learn endodontic access cavity preparations [28]. Notwithstanding the initial cost, use of VRDS presents several advantages including the option to repeat the exercises multiple times without any expenditure on materials or fear of making an error. Training of VRDS is environmentally friendly as it does not produce any waste. Additionally, there are options to build additional cases in the VRDS library to enhance student training including simulation of real patient cases [29–31].

The present study focused on the learning experiences of undergraduate students in endodontic access cavity preparations using conventional models as well as VRDS at two institutions. Although different VRDS systems were used by the participating institutions in this study, both systems incorporated a touchscreen, dental handpiece, space mouse, dental mirror, and speed pedal. The virtual models of both systems also simulated dental tissues (enamel, dentin, pulp) pulp chamber, pulp floor, as well as dental caries.

This study explores the impact of VRDS on endodontic access cavity preparations in undergraduate dental education programs. A vast majority of the participants shared positive perceptions and experiences with VRDS training suggesting that it can certainly complement pre-clinical training using acrylic and natural teeth. However, in its present form, VRDS cannot replace training on conventional dental models, which simulate clinical endodontics on real patients more closely. These findings corroborate previous studies [12,32,33]. However, in contrast to previous study [4], a high proportion of the participants felt that access cavity preparation on VRDS closely matched the same on natural as well as acrylic teeth. The virtual tooth closely simulated the differential hardness of enamel, dentine, and pulp, which allowed participants to feel tooth cutting and drop in the pulp chamber in a comparable way as experienced on natural and acrylic teeth. These differences may, at least in part, reflect improvements in haptic feedback provided by VRDS equipment as some of the previous studies reporting lower satisfaction with the quality haptic feedback were conducted several years ago.

The participants in this study provided a few recommendations to further improve the VRDS technology including improvements in software to capture variations in root canal anatomy, options to negotiation root canal orifices, and practice irrigation. Improvements in the ergonomic design of the VRDS equipment were also suggested for better access, vision and improved control of the virtual handpiece and dental mirror. Dental students represent the key stakeholders in dental education and their feedback can be fed back to the manufacturers to further enhance the VRDS technology.

VRDS was perceived to provide a less stressful learning environment for novice students as it allows correction of errors and deficiencies without the need to replace the models. The use of a virtual handpiece offers a low-risk learning environment compared to training on models involving the use of a physical handpiece. Moreover, minimal running expenses of VRDS make it a useful learning tool to introduce endodontic procedures before the students start consolidating their skills on dental models. Participants in this study expressed the need to have additional opportunities for training on VRDS and suggested that in addition to the automated feedback provided by the software, additional advice from supervisors would be helpful for the students. A recent study also supports the use of VRDS to provide clinically relevant and qualitative feedback to undergraduate dental students [34]. A critical review on the use of VR in dentistry also emphasizes timely feedback and deliberate practice to translate the benefits of VRDS into competence of dental students [35]. These recommendations are in accordance with the findings of the present study. Based on these findings, we aim to provide a more structured preclinical training on VRDS to our future cohorts at an earlier stage of the undergraduate dental curriculum. Such

an approach may allow more time for the development of core skills before they translate them on physical models and real patients in preclinical and clinical settings, respectively.

A few limitations of the study need to be mentioned. Firstly, the sample size of this study was small due to the size of the study cohorts. Future studies involving a larger sample size from multiple institutions may serve to enhance the generalizability of the findings. Secondly, the intervention evaluated the experience of students on VRDS involving a single exercise in pre-clinical settings. In addition to standard models available in the VRDS library, it might be helpful to provide additional training on simulated natural teeth. Moreover, longitudinal follow-up of students in their transition to clinical settings, and triangulation of their performance on real patients may allow educators to capture a more holistic picture of their competency development in endodontic access cavity preparation. Nevertheless, the findings of this study provide useful insights into the strengths and limitations of VRDS training in endodontic access cavity preparation and recommendations to further improve the learning experiences of students.

## 5. Conclusions

The study provides useful insights into the strengths and limitations of VRDS in endodontics training of undergraduate dental students. The findings of this study support the use of VRDS to complement conventional methods of preclinical training in simulated laboratory settings. The participants made useful recommendations to improve VRDS technology, which may enhance the learning experiences of students in the future. Dental educators must work closely with the manufacturers to optimize the VRDS in undergraduate dental education.

**Author Contributions:** R.B.-H., D.H.-Y. and K.A. conceptualized the study, B.C., K.O., H.D., L.A., T.A., M.R., R.B.-H., D.H.-Y. and S.A. contributed to data collection, R.G. performed the data analysis, R.B.-H., D.H.-Y., L.A., T.A. and K.A. wrote the original draft, K.A. reviewed and drafted the final manuscript. All authors have read and agreed to the published version of the manuscript.

**Funding:** This research received no external funding.

**Institutional Review Board Statement:** The study was conducted in accordance with the Declaration of Helsinki, and approved by the Institutional Review Board the Qatar University (Protocol code: QU-IRB 1652-EA/22).

**Informed Consent Statement:** An informed consent was obtained from all the participants.

**Data Availability Statement:** The data that support the findings of this study are available on request from the corresponding author.

**Conflicts of Interest:** The authors declare no conflict of interest.

## Appendix A

**Table A1.** Count of responses.

| Item | Institution | Strongly Agree | Agree | Unsure | Disagree | Strongly Disagree |
|---|---|---|---|---|---|---|
| 1. The case presentation on VR allowed me to clearly comprehend the tasks expected from me | | | | | | |
| 2. The hardness and texture of models in the VR felt similar to natural teeth | | | | | | |
| 3. The hardness and texture of models in the VR felt similar to artificial teeth used for endodontic training | | | | | | |

**Table A1.** *Cont.*

| Item | Institution | Strongly Agree | Agree | Unsure | Disagree | Strongly Disagree |
|---|---|---|---|---|---|---|
| 4. The images of the teeth and instruments on the VR accurately simulated real structures | | | | | | |
| 5. Access cavity preparation on VR was easier compared to acrylic teeth | | | | | | |
| 6. The training exercise on the VR improved my fine motor skills | | | | | | |
| 7. VR training improved my confidence in learning Endodontics | | | | | | |
| 8. VR can replace the preclinical endodontics training on models | | | | | | |
| 9. I would like to have more VR sessions in Endodontics | | | | | | |
| 10. VR Training may be used to supplement pre-clinical training on mannequin models | | | | | | |

Open-ended questions

1. In your opinion what are the main advantages of learning Endodontics on VR?
2. What are the main limitations of learning Endodontics on VR?
3. Did you experience any difficulties in using VR Hardware or Software? Please explain
4. Is the automated feedback provided by VR adequate or would additional feedback by the facilitator be essential? Please explain

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
