# Peer review of "Impact of Virtual Reality Simulation in Endodontics on the Learning Experiences of Undergraduate Dental Students"

_applsci, doi:10.3390/app13020981_

Round 1

Reviewer 1 Report

Very good topic of study. It would only be necessary to increase the explanation on how the improvement of learning will be evidenced, in an explicit and more extensive way. 

It is suggested that the presentation of results in the summary be described in a general and not specific way in order to know them in the results section in greater detail. 

It is hoped that they can increase the information of the conceptual framework regarding virtual reality for teaching in dentistry and in the learning sector indicated in the article.

Point out what the competency to prepare cavities in students is about, what are the skills.  

Explain the questions asked for sample selection.

Increase the quotations from five years ago, considering strengthening the conceptual framework presented. 

Author Response

Dear Editor,

The authors would like to thank the editorial team and the reviewers for their constructive feedback which are very helpful to improve the quality and rigor of the manuscript. The authors can confirm that we have addressed the comments meticulously. The corrections are highlighted in red in the revised manuscript and a parawise response to reviewers’ feedback is provided below:

Reviewer 1:

Comment 1: Very good topic of study. It would only be necessary to increase the explanation on how the improvement of learning will be evidenced, in an explicit and more extensive way.

Response: Amended in the manuscript as suggested. 

Comment 2: It is suggested that the presentation of results in the summary be described in a general and not specific way in order to know them in the results section in greater detail.

Response: Amended in the revised manuscript as suggested.

 Comment 3: It is hoped that they can increase the information of the conceptual framework regarding virtual reality for teaching in dentistry and in the learning sector indicated in the article.

Response: The information linking the conceptual framework with the virtual reality teaching in dentistry and more widely the higher education has been revised to make it more clear and comprehensive.

 Comment 4: Point out what the competency to prepare cavities in students is about, what are the skills. 

Response: Additional details regarding access cavity competency have been added in the revised manuscript as suggested.

 Comment 5: Explain the questions asked for sample selection.

Response: Additional details regarding the sample selection have been added in the revised manuscript as suggested.

 Comment 6: Increase the quotations from five years ago, considering strengthening the conceptual framework presented.

Response: Amendments made in the revised manuscript as suggested.

Reviewer 2 Report

The abstract said that 'the sample included 60 year - 3 dental undergraduate students -- it is not a clear sentence; what does it mean?

The introduction could be improved; with the background of the research and the purpose of the study --investigate the learning experience towards VR Training, the key research question is about the value of pre-clinical VR Training in one of the courses in curricula. --- the connection between the learning experience with curricula using VR could be improved, in line with the method of research and the result. 

The methodologies must be improved; what are the research questions, and followed with the design and appropriate sample technique. Is it possible to use a purposive sample? why not use probability sampling? since I saw the construct of research related to quantitative research and using the number of students from two universities. 

The result must be described in communicative ways; about the table please inform us more clearly what it means. 

The discussion must be improved by the result of the research. 

Author Response

Dear Editor,

The authors would like to thank the editorial team and the reviewers for their constructive feedback which are very helpful to improve the quality and rigor of the manuscript. The authors can confirm that we have addressed the comments meticulously. The corrections are highlighted in red in the revised manuscript and a parawise response to reviewers’ feedback is provided below:

Reviewer 2:

Comment 1: The abstract said that 'the sample included 60 year - 3 dental undergraduate students -- it is not a clear sentence; what does it mean?

Response: This sentence has been revised to improve clarity.

Comment 2: The introduction could be improved; with the background of the research and the purpose of the study --investigate the learning experience towards VR Training, the key research question is about the value of pre-clinical VR Training in one of the courses in curricula. --- the connection between the learning experience with curricula using VR could be improved, in line with the method of research and the result.

Response: The introduction section has been amended in the revised manuscript as suggested.

 Comment 3: The methodologies must be improved; what are the research questions, and followed with the design and appropriate sample technique. Is it possible to use a purposive sample? why not use probability sampling? since I saw the construct of research related to quantitative research and using the number of students from two universities.

Response: The methodology section including the research question and design etc. have been amended to enhance clarity. Regarding the sampling technique, purposive sampling (a group of non-probability sampling) was used as it allows selection of participants with specific characteristics i.e., undergraduate dental students who have experienced virtual reality and conventional mannequin training in endodontics in a single course in Endodontics.

Comment 3: The result must be described in communicative ways; about the table please inform us more clearly what it means.

Response: Further explanation of the Information in the tables has been added in the revised manuscript as suggested.

 Comment 4: The discussion must be improved by the result of the research.

Response: The discussion section has been revised with further exploration of application of virtual reality in undergraduate dental education and is supported by additional references.

Reviewer 3 Report

The topic is able to bring attention to readers. It provided background information with updated literature. The graphical illustration is clearly presented.

Few concerns before moving to publication:

·       Recommend to fine-tune the research topic and include the information where the research takes place (ie Qatar and Turkey)

·       The research question could be more specific.

·       More examples around the globe could be taken into reference for exploring further

·       Recommend to increase the sample size.

·       Try to provide a more in-depth analysis with the support of the data collection

Author Response

Dear Editor,

The authors would like to thank the editorial team and the reviewers for their constructive feedback which are very helpful to improve the quality and rigor of the manuscript. The authors can confirm that we have addressed the comments meticulously. The corrections are highlighted in red in the revised manuscript and a parawise response to reviewers’ feedback is provided below:

Reviewer 3:

The topic is able to bring attention to readers. It provided background information with updated literature. The graphical illustration is clearly presented.

Response: Thank you

Few concerns before moving to publication:

Comment 1:  Recommend to fine-tune the research topic and include the information where the research takes place (ie Qatar and Turkey)

Response: Amended in the revised manuscript as suggested with information re the setting of the research i.e., Qatar and Turkey have been added...

Comment 2: The research question could be more specific.

Response: The research question has been amended in the revised draft as suggested.

Comment 3: More examples around the globe could be taken into reference for exploring further

Response: Additional examples of application of virtual reality in dental education have been added and are supported by citations

 Comment 4: Recommend to increase the sample size.

Response:

The sample size is dependent on the strength of the participating cohorts and cannot be increased. It is recognized that the sample size is low and therefore, it is acknowledged as a limitation of the study in the discussion section.

Comment 5: Try to provide a more in-depth analysis with the support of the data collection

Response: Amended in the manuscript as suggested.

Round 2

Reviewer 3 Report

Thanks for your effort in revising the paper. It provides better clarity of this research to the readers.

For the aim of the study, please check the writing and consider to re-phrase to either questions or statements.

The aim of this study was to explore the value of VRDS in pre-clinical training of undergraduate dental students in endodontics and evaluate how it impacts on their learning experiences? 

Author Response

Dear Editor,

The authors can confirm that we have addressed reviewer comments on the first revision. The corrections are highlighted in red in the revised manuscript and a parawise response to reviewers’ feedback is provided below:

Reviewer 3:

Thanks for your effort in revising the paper. It provides better clarity of this research to the readers.

For the aim of the study, please check the writing and consider to re-phrase to either questions or statements.

The aim of this study was to explore the value of VRDS in pre-clinical training of undergraduate dental students in endodontics and evaluate how it impacts on their learning experiences? 

Response: Thank you

The aim of the study has been amended